# Contrastive multiple correspondence analysis (cMCA): Using contrastive learning to identify latent subgroups in political parties

Takanori Fujiwara [1], Tzu-Ping Liu [2]*

1 Linköping University, Norrköping, Sweden, 2 University of Taipei, Taipei, Taiwan

☯ These authors contributed equally to this work.
* tpliu@utaipei.edu.tw

## Abstract

Scaling methods have long been utilized to simplify and cluster high-dimensional data. However, the *general* latent spaces *across* all predefined groups derived from these methods sometimes do not fall into researchers' interest regarding *specific* patterns *within* groups. To tackle this issue, we adopt an emerging analysis approach called contrastive learning. We contribute to this growing field by extending its ideas to multiple correspondence analysis (MCA) in order to enable an analysis of data often encountered by social scientists—containing binary, ordinal, and nominal variables. We demonstrate the utility of contrastive MCA (cMCA) by analyzing two different surveys of voters in the U.S. and U.K. Our results suggest that, first, cMCA can identify substantively important dimensions and divisions among subgroups that are overlooked by traditional methods; second, for other cases, cMCA can derive latent traits that emphasize subgroups seen moderately in those derived by traditional methods.

## Introduction

Scaling has been a prolific and popular topic throughout political and social science. Scholars have developed and utilized a diverse set of methods to uncover similarities/differences among political actors in a latent space. This has been accomplished by utilizing one of two general ideas: accumulative and unfolding models [1–3]. The widely utilized accumulative model is the classic item response theory (IRT) model, which re-scales the data by arranging a series of questions in order of their level of discrimination regarding the measurement of certain latent traits [1]. On the contrary, unfolding models (e.g., the NOMINATE class of models and Optimal Classification (OC)) assume that each individual has an ideal point and this individual's preference is modeled by a single-peaked function, which translates proximity information into distances in a latent space [2, 4]. While these methods have been most frequently applied to roll-call votes in the United States Congress or other political bodies, scaling methods that do not belong to the above categories such as PCA, MCA, black-box scaling, and ordinal IRT (OIRT) have been increasingly used to analyze a range of new data, especially surveys, to explore underlying political patterns among citizens.

**Data Availability Statement:** Data are located at: https://github.com/tzuliu/Contrastive-Multiple-Correspondence-Analysis-cMCA.

**Funding:** This study was supported by the Taiwan National Science and Technology Council in the

form of a grant to T-PL [110-2410-H-845 -030 -MY2] and by The Knut and Alice Wallenberg Foundation in the form of a grant to TF [KAW 2019.0024].

**Competing interests:** The authors have declared that no competing interests exist.

Ever since the work by Downs [5], the spatial modelers have theoretically and empirically posited the existence of a single-dimensional ideological space [6–8], and the latent left-right dimension of ideology is often recovered as an important principal component (PC) in political data [9–11]. Indeed, almost all of the above methods recover ideology as their first PC in the derived latent space. These lower-dimensional representations can both help us understand the *general* structure of a given dataset, such as its distribution, and identify underlying patterns, such as clusters and outliers [12, 13]. However, as in the aforementioned ideology PC, the general structure and related patterns may not be particularly interesting or important to researchers. Rather, using alternatives, researchers may be interested in exploring *specific structure and patterns within each of groups* (e.g., citizens supporting different political parties).

A variation that is overlooked *across* groups may be significant *within* groups, as seen in substantive applications in political science. Indeed, traditional scaling methods sometimes only uncover certain intraparty divisions that are related to the "mainstream" pattern (e.g., when applying scaling methods only to Democrats or Republicans, the explored subgroups are usually consistent with ideological differences). This demonstrates how traditional methods may merely concentrate on the general patterns among *all* actors but then necessarily do not identify latent trends within groups and their corresponding causes, such as factors that divide a group into subgroups. It is important to note that we by no means consider existing models to be problematic or inferior—we expect an alternative approach that focuses on variation specific within each group to be especially fruitful in cases where subgroups may lurk in the data and be overlooked by ordinary approaches.

In this article, we use contrastive learning to analyze potential differences above and beyond these general patterns. While these mentioned scaling methods analyze the data as a whole, contrastive scaling instead *contrasts* two groups within the data to utilize their differences to find intra-group patterns. The idea of "contrastive scaling" is straightforward—through comparison of two groups, contrastive scaling identifies principle directions/components on which the data of one group varies largely but only slightly for the data of the other group [14]. Practically, contrastive scaling first splits data into different groups, usually by predefined classes (e.g., party ID), and then compares the data of the target group against the background group to identify PC(s) on which *the target group has relatively high variance* and *the background group has relatively low variance*.

Compared to the most prolific contrastive scaling method—contrastive PCA (cPCA) [14]—we contribute to the field of contrastive learning by extending cPCA's idea from PCA to MCA to develop contrastive MCA (cMCA). cMCA allows researchers to apply contrastive scaling to noncontinuous data—binary, ordinal, and nominal data—that is encountered frequently but cannot be analyzed by cPCA properly. With cMCA, researchers can keep the completeness of data as much as possible while analyzing the survey with contrastive scaling (i.e., without dropping noncontinuous data). This is important as listwise deletion (due to variable-type cannot be analyzed) can bias the distribution and further cause false recovery. As a member of the contrastive scaling family, cMCA first splits the data into predefined groups, such as partisanship or treatment and control groups; second, as in the case with MCA, cMCA applies a one-hot encoder to convert a categorical dataset into a binary-format matrix, called a disjunctive matrix; finally, cMCA takes one of the groups as the target and then compares this group with the background group to derive PCs on which the positions of members' ideal points from the target group vary the most but those from the background group vary the least.

To demonstrate the utility of cMCA, we analyze two citizen surveys, the 2020 Cooperative Election Survey (CES 2020), which investigates general political attitudes, various

demographic factors, assessment of roll call voting choices, political information, and vote intentions of American voters, and the U.K. module of the 2018 European Social Survey (ESS-UK 2018), which is a biennial cross-national survey of attitudes and behaviors across member countries. The results show that other than the general pattern among all the respondents, which is derived through traditional scaling, cMCA *objectively* detects subgroups along with certain directions that are overlooked by traditional approaches. By objectivity, we mean that instead of picking variables/attributes based on researchers' own (subjective) prior beliefs regarding the cause of intra-polarization, cMCA (or contrastive learning in general) explores these attributes by recognized statistical properties/criteria, e.g., variables which cause the largest variance to the distribution. Such subgroups include those corresponding to the attitudes on Trump's two recent Supreme Court nominations and on Trump's job approval among Republicans in the U.S.; the pro- and anti-EU attitudes among supporters from each of the Conservative and Labour parties in the U.K. Regardless of whether traditional scaling finds a clear pattern among the respondents or not, cMCA derives its own salient factors to identify subgroups from a specific group. This article makes an important contribution to *scaling methods*, *unsupervised learning*, *data mining*, and *visualization* for complex data by enabling researchers to identify latent dimensions that effectively classify and divide observations (voters) even when the left-right scale cannot effectively discriminate between classes/groups (partisanship). Note that although cMCA compares predefined groups first, it is still considered unsupervised learning given that the labels which categorize/cluster samples within each predefined group remain *unknown* before conducting the analysis. Simply put, cMCA's results provide important information for researchers to better understand and make distinctions between underlying subgroups in their data.

Finally, to date, almost all of the methods for subgroup analysis, such as class specific MCA (CSA) [15, 16] and subgroup MCA (sMCA) [17] (see Discussion for detailed discussion), require researchers to already know how data should be "subgrouped" in the latent dimensions (i.e., what variables may cause divisions *within* groups), and therefore, without such prior knowledge, these methods are almost infeasible to use, especially when data is high-dimensional. By revealing the overlooked influential factors, cMCA indeed provides a springboard, i.e., feature extraction, for further subgroup analyses. The paper proceeds with a description of cMCA and its application to the two voter surveys; then, through the comparison with current subgroup analysis methods, we conclude the paper with general thoughts about and rules of thumb for using cMCA as well as its application to substantive topics.

## Contrastive learning and contrastive MCA

Contrastive learning is an emerging machine learning approach that analyzes high-dimensional data to capture "patterns that are specific to, or enriched in, one data relative to another" [14]. Unlike ordinary scaling methods, which usually define PCs along which data as a whole has the largest variance or best demonstrates the (dis)similarity between observations in the data, the logic behind contrastive scaling is to instead define principal components/directions that better capture the distribution of data within one group (the target group) in *contrast* to another (the background group). Specifically, contrastive scaling identifies PCs that capture the *most* variance within the target data and the *least* variance in the background data. So far, contrastive learning has been applied to several machine learning methods, including PCA [14], latent Dirichlet allocation [18], hidden Markov models [18], regressions [19], and variational autoencoders [20]. We utilize this contrastive learning approach by applying it to MCA, an enhanced version of PCA for nominal and ordinal data analysis [16, 21]. MCA is a valuable tool for exploring principle components of categorical data; however, it has not been widely

utilized in political science (see [22–24] for exceptions). The source code of the derived method, contrastive MCA (cMCA), is available on our online repository: https://github.com/takanori-fujiwara/cmca.

## Multiple correspondence analysis (MCA)—an alternative to PCA

Although PCA has been widely used across disciplines for scaling, dimensional reduction, and data visualization, this approach has major limitations that may cause issues in analytical results. That is, as PCA assumes the relationships between variables are *linear* [25], PCA does not effectively analyze categorical data [26]—including binary data, nominal data, and ordinal data, where the parallel slopes assumption does not often hold [27]. One of the standard practices to resolve this issue is to create dummy variables for all categorical data, then standardize the whole data, and finally apply PCA to this new dataset—this procedure is commonly recognized as a variant of PCA for categorical data and referred to as *multiple correspondence analysis* (MCA) by some scholars [21].

With similar analytical procedures to PCA, as an alternative, MCA can include all types of noncontinuous variables in the analysis to overcome the above issues. Practically, given that researchers frequently transform continuous variables to ordinal variables while analyzing surveys, such as age, years spent at school, and so on, MCA does not necessarily exclude continuous variables from the analysis. During the analysis, MCA first converts an input noncontinuous dataset, $\mathbf{X} \in \mathbb{R}^{p \times d}$ (*p*: the number of data points, *d*: the number of variables), into what is called a disjunctive matrix, $\mathbf{G} \in \mathbb{R}^{p \times K}$ (*K*: the total number of categories), by applying one-hot encoding to each of *d* categorical dimensions [28]. For illustration, assume that $\mathbf{X}$ consists of two columns/variables, `color` and `shape`, and the variables have three (`red`, `green`, `blue`) and two (`circle`, `rectangle`) levels, respectively. In this case, the disjunctive matrix $\mathbf{G}$ will contain five categories: `red`, `green`, `blue`, `circle`, and `rectangle`. For instance, a piece of blue rectangle paper will be represented as a row vector, [0, 0, 1, 0, 1], in $\mathbf{G}$.

We can further derive a probability/correspondence matrix, $\mathbf{Z}$, through dividing each value of $\mathbf{G}$ by the grand total of $\mathbf{G}$ (i.e., $\mathbf{Z} = N^{-1}\mathbf{G}$ where *N* is the grand total of $\mathbf{G}$). This probability matrix can now be treated as a typical dataset for use in continuous data analyses. Given that the probability can be treated as a type of data, similar to PCA, we first normalize $\mathbf{Z}$, resulting $\mathbf{Z}_n$; then, obtain what is called a Burt matrix, $\mathbf{B}$, with $\mathbf{B} \stackrel{\text{def}}{=} \mathbf{Z}_n^{\top}\mathbf{Z}_n$ ($\mathbf{B} \in \mathbb{R}^{K \times K}$). This Burt matrix $\mathbf{B}$ under MCA corresponds to a variance-covariance matrix under PCA. Thus, as in PCA, to derive principal directions, MCA performs eigenvalue decomposition (EVD) to $\mathbf{B}$ to preserve the variance of $\mathbf{G}$. Similar to PCA, without computing $\mathbf{B}$, one could directly apply singular value decomposition (SVD) to $\mathbf{Z}_n$ to obtain the same principal directions.

## Extending MCA to cMCA

To date, there is only one application of contrastive learning to a scaling method—cPCA [14] and its variants (e.g., online cPCA [29], sparse cPCA [30], and unified linear comparative analysis [31]). Given that the procedure of cPCA adds only one additional step into PCA to compare two groups, cPCA and its variants inevitably inherit PCA's limitations. Such limitations impede the usage of contrastive scaling to survey data. To apply contrastive learning to MCA as an alternative to cPCA, we first split the original dataset, $\mathbf{X} \in \mathbb{R}^{p \times d}$, into a target group, $\mathbf{X}_T \in \mathbb{R}^{n \times d}$, and a background group, $\mathbf{X}_B \in \mathbb{R}^{m \times d}$ (where $p = n + m$), by a certain predefined boundary, such as individuals' partisanship or whether they were assigned to the treatment or control groups. We then further derive two Burt matrices from the target and background

groups, $\mathbf{B}_T$ and $\mathbf{B}_B$, respectively. Importantly, because the contrastive learning procedure utilizes *matrix subtraction*, the Burt matrices must have the same dimensions, i.e., $\mathbf{B}_T \in \mathbb{R}^{K \times K}$ and $\mathbf{B}_B \in \mathbb{R}^{K \times K}$. Note that this does not mean that the original datasets of two groups must have the same sample size, as expressed as $\mathbf{X}_T \in \mathbb{R}^{n \times d}$ and $\mathbf{X}_B \in \mathbb{R}^{m \times d}$ (on the other hand, the original datasets of two groups must have the same set of $d$ variables). Both cPCA and cMCA only compare the variance-covariance or Burt matrices of two original datasets. When two datasets have the same set of variables, their variance-covariance/Burt matrices are guaranteed to have the same dimensions.

Let $\mathbf{u}$ be any $K$-dimensional vector. Similar to cPCA, we can derive the variances of the target and background groups along $\mathbf{u}$ with $\sigma_T^2(\mathbf{u}) \overset{\text{def}}{=} \mathbf{u}^\top \mathbf{B}_T \mathbf{u}$ and $\sigma_B^2(\mathbf{u}) \overset{\text{def}}{=} \mathbf{u}^\top \mathbf{B}_B \mathbf{u}$. In addition, since we want to find a "direction" (i.e., we are not interested in the length of $\mathbf{u}$), we impose a constraint that sets the derived direction to be a *unit vector*, i.e., $\|\mathbf{u}\| = 1$. Let $\Sigma$ denote $(\mathbf{B}_T - \alpha \mathbf{B}_B)$. To find the direction, $\mathbf{u}^*$, on which a target group's probability matrix, $\mathbf{Z}_T$, has a large variance and a background group's probability matrix, $\mathbf{Z}_B$, has a small variance, one needs to solve the optimization problem:

$$\mathbf{u}^* = \arg\max_{\mathbf{u}} \; \sigma_T^2(\mathbf{u}) - \alpha\sigma_B^2(\mathbf{u}) = \arg\max_{\mathbf{u}} \; \mathbf{u}^\top(\mathbf{B}_T - \alpha\mathbf{B}_B)\mathbf{u} \tag{1}$$

where $\alpha$ ($0 \leq \alpha \leq \infty$) is a hyperparameter of cMCA, called the *contrast parameter*. From Eq 1, because both $\mathbf{B}_T$ and $\alpha\mathbf{B}_B$ are symmetric matrices, $(\mathbf{B}_T - \alpha\mathbf{B}_B)$ is also symmetric as the transpose of $\mathbf{B}_T - \alpha\mathbf{B}_B$ is equal to itself, i.e., $(\mathbf{B}_T - \alpha\mathbf{B}_B)^\top = \mathbf{B}_T^\top - \alpha\mathbf{B}_B^\top = \mathbf{B}_T - \alpha\mathbf{B}_B$. In other words, we can simply apply the analytical procedure of MCA to this matrix, $(\mathbf{B}_T - \alpha\mathbf{B}_B)$.

Let $\Sigma$ denote $(\mathbf{B}_T - \alpha\mathbf{B}_B)$, and we now rewrite Eq 1 as follow:

$$\mathbf{u}^* = \arg\max_{\|\mathbf{u}\|=1} \; \mathbf{u}^\top\Sigma\mathbf{u} \tag{2}$$

This maximization problem can be solved by using *Lagrangian function* which is transformed from Eq 2 by the method of *Lagrange multiplier*:

$$\mathcal{L}(\mathbf{u}, \lambda) = \mathbf{u}^\top\Sigma\mathbf{u} - \lambda(\mathbf{u}^\top\mathbf{u} - 1) \tag{3}$$

Taking the derivative on Eq 3 with respect to $\mathbf{u}$, we have the following equation:

$$\nabla_{\mathbf{u}}\mathcal{L} = \Sigma\mathbf{u} - \lambda\mathbf{u} \tag{4}$$

By setting Eq 4 to be equal to 0, we finally derive that $\Sigma\,\mathbf{u} = \lambda\mathbf{u}$, which implies that $\lambda$ and $\mathbf{u}$ are the eigenvalue and eigenvector of $\Sigma$, respectively [32]. In other words, $\mathbf{u}^*$ corresponds to the first eigenvector of the matrix $(\mathbf{B}_T - \alpha\mathbf{B}_B)$ and can be derived through performing EVD over $(\mathbf{B}_T - \alpha\mathbf{B}_B)$. Unlike MCA, SVD cannot be directly applied to obtain eigenvectors for cMCA because we need to compute the difference between target and background groups' Burt matrices (i.e., $\mathbf{B}_T - \alpha\mathbf{B}_B$). Note that the same procedure can be applied to derive a set of multiple eigenvectors/directions, $\mathbf{U} \in \mathbb{R}^{K \times K'}$, where $K'$ is the number of the derived top eigenvectors.

## Selection of the contrast parameter

The contrast parameter $\alpha$ in Eq 1 controls the trade-off between having high target variance versus low background variance. When $\alpha = 0$, cMCA only maximizes the variance of a target group, and produces results that are equivalent to applying standard MCA only to the target group. As $\alpha$ increases, cMCA places a greater emphasis on directions that reduce the variance of a background group. As proved by [33], arbitrarily selected $\alpha$ values within $0 \leq \alpha \leq \infty$ can

produce different latent spaces, each of which shows a different ratio of high target variance to low background variance. Thus, researchers can manually explore different $\alpha$ values to seek latent spaces that demonstrate clear patterns of data division/clustering for them. In other words, by examining various different $\alpha$ values in cMCA, researchers can investigate multiple potential factors that reveal unique patterns in the target group relative to the background group.

In addition to manual selection, we extend the automatic selection of a value of $\alpha$ for cPCA [34] to the one for cMCA, with which researchers can find a value of $\alpha$ that derives a latent space where a target group has the *highest* variance relative to a background group's variance. Given that $\mathbf{U}^\top \mathbf{B}_T \mathbf{U}$ and $\mathbf{U}^\top \mathbf{B}_B \mathbf{U}$ are respectively target and background groups' variance-covariance matrices under a latent space, the sum of the diagonal of each variance-covariance matrix, i.e., $\mathrm{tr}(\mathbf{U}^\top \mathbf{B}_T \mathbf{U})$ and $\mathrm{tr}(\mathbf{U}^\top \mathbf{B}_B \mathbf{U})$, represents the total variance of each group. Thus, this specific $\alpha$ can be found by solving the following trace-ratio problem:

$$\max_{\mathbf{U}^\top \mathbf{U} = \mathbf{I}} \frac{\mathrm{tr}(\mathbf{U}^\top \mathbf{B}_T \mathbf{U})}{\mathrm{tr}(\mathbf{U}^\top \mathbf{B}_B \mathbf{U})} \tag{5}$$

Eq 5 finds the highest ratio between the total variances of the target and background groups and treats this ratio as the desired contrast parameter, $\alpha$, to derive the corresponding eigenvectors and latent spaces. Following the work by Dinkelbach [35], we employ an iterative algorithm to solve Eq 5 as directly finding a solution is usually difficult. The iterative algorithm consists of two steps: Given eigenvectors, $\mathbf{U}_s$, at iteration step $s$ ($s \geq 0$ and $s \in \mathbb{Z}$), we perform

$$\textbf{Step1.} \qquad \alpha_s \leftarrow \frac{\mathrm{tr}(\mathbf{U}_s^\top \mathbf{B}_T \mathbf{U}_s)}{\mathrm{tr}(\mathbf{U}_s^\top \mathbf{B}_B \mathbf{U}_s)} \tag{6}$$

$$\textbf{Step2.} \qquad \mathbf{U}_{s+1} \leftarrow \arg \max_{\mathbf{U}^\top \mathbf{U} = \mathbf{I}} \mathrm{tr}(\mathbf{U}^\top (\mathbf{B}_T - \alpha_s \mathbf{B}_B)\mathbf{U}) \tag{7}$$

At the beginning of the iteration (i.e., $s = 0$), since the computed $\mathbf{U}_0$ does not exist, we self-define $\alpha_0 = 0$ as the default solution to Step 1. As demonstrated, $\alpha_s$ in Eq 6 is an objective value of Eq 5, which is computed with the current $\mathbf{U}_s$. The second step (Eq 7) is to derive the eigenvectors, $\mathbf{U}_{s+1}$, for the next iteration. This step just solves the original cMCA problem based on the current contrast parameter, $\alpha_s$. With this iterative algorithm, $\alpha_s$ monotonically increases to a value that satisfies Eq 5 and usually converges quickly (i.e., in less than 10 iterations). For detailed mathematical explanations, please refer to [35].

Note that when $\mathbf{B}_B$ is nearly singular, $\alpha_s$ approaches infinity. One potential solution to avoid this issue is adding a small constant value, $\varepsilon$ (e.g., $\varepsilon = 10^{-3}$ by default), to each diagonal element of $\mathbf{B}_B$. However, $\varepsilon$ does not have a clear connection to the optimization problem in Eq 5, and consequently, it is hard for researchers to control $\alpha$'s search space. Therefore, we add $\varepsilon \mathrm{tr}(\mathbf{U}^\top \mathbf{B}_T \mathbf{U})$ to the denominator of Eq 6, i.e., $\alpha_s \leftarrow \frac{\mathrm{tr}(\mathbf{U}_s^\top \mathbf{B}_T \mathbf{U}_s)}{\mathrm{tr}(\mathbf{U}_s^\top \mathbf{B}_B \mathbf{U}_s) + \varepsilon\, \mathrm{tr}(\mathbf{U}_s^\top \mathbf{B}_T \mathbf{U}_s)}$. This allows us to control the search space so that $\alpha_s$ reaches at most $1/\varepsilon$ (e.g., when $\varepsilon = 10^{-3}$, $\alpha_s$ may increase up to 1,000).

## Data-point coordinates, category coordinates, and category loadings

As in ordinary MCA, in cMCA, we provide three essential tools to help researchers relate data points and categories/variables to the contrastive latent space: (1) data-point coordinates (also known as coordinates of rows [21] or clouds of individuals [16]), (2) category coordinates (also known as coordinates of columns [21] or clouds of categories [16]), and (3) category loadings.

Data-point coordinates provide a lower-dimensional representation of the data, which is standard in most dimensional reduction techniques. On the contrary, category coordinates and loadings provide essential information on how to interpret these data-point coordinates. Similar to data-point coordinates, category coordinates present the position of each category/level in a lower-dimensional space. Given that the coordinates of each category are on the same contrastive latent space with data-point coordinates (see Eq 10), by comparing these two sets of coordinates, we can better understand the associations between the data points and categories. Respondents who are placed close to the position of some category are highly likely to hold this category for the corresponding variable [16, 21, 36]. On the other hand, category loadings indicate how strongly each category contributes to each derived eigenvector.

Similar to MCA, for a target group, $\mathbf{X}_T$, whose probability matrix is $\mathbf{Z}_T$, cMCA's data-point coordinates, $\mathbf{Y}_T^{\text{row}} \in \mathbb{R}^{n \times K'}$ ($K' < K$), can be derived as:

$$\mathbf{Y}_T^{\text{row}} = \mathbf{Z}_T \mathbf{U} \tag{8}$$

where $\mathbf{U} \in \mathbb{R}^{K \times K'}$ is the top-$K'$ eigenvectors obtained by EVD as demontned in Eq 1. In other words, $\mathbf{Y}_T^{\text{row}}$ is $\mathbf{Z}_T$'s projected positions onto the new space defined with $\mathbf{U}$. We call this new space's axes (i.e., linear combinations of the original categories and the eigenvectors) *contrastive principal components* (cPCs). Although the eigenvectors, $\mathbf{U}$, are sometimes called cPCs in existing literature (e.g., [12, 14]), it is inaccurate according to Jolliffe and Cadima [37]. Similarly, we can obtain data-point coordinates, $\mathbf{Y}_B^{\text{row}}$, of a background group, $\mathbf{X}_B$, (its probability matrix is $\mathbf{Z}_B$) onto the same low-dimensional space with $\mathbf{Y}_T^{\text{row}}$ through:

$$\mathbf{Y}_B^{\text{row}} = \mathbf{Z}_B \mathbf{U} \tag{9}$$

As discussed in [38], visualizing low-dimensional representations of both target and background groups can help researchers determine whether or not a target group has certain unique patterns relative to a background group. When the scatteredness/shape of plotted data points of $\mathbf{Y}_T^{\text{row}}$ is *greatly larger* than $\mathbf{Y}_B^{\text{row}}$, we can conclude that $\mathbf{Y}_T^{\text{row}}$ contains the unique patterns.

To derive the target group's category coordinates, $\mathbf{Y}_T^{\text{col}}$, we follow a way taken in MCA:

$$\mathbf{Y}_T^{\text{col}} = \mathbf{D}_T^{-1/2} \mathbf{U} \tag{10}$$

where $\mathbf{D}_T$ is a diagonal matrix that has the sum of each column of the standardized probability matrix of a target group (this sum is conventionally called *column mass* [21]), as each diagonal element. One can consider Eq 10 similar to PCA's standardized variable coordinates—instead of the Euclidean distance used in PCA, MCA or cMCA adopts $\chi^2$ distance under the latent space (see [21, 39] for detailed discussion).

Finally, since MCA's derivation procedure is highly similar to that of PCA, we utilize the concept of normalized PC loadings in PCA to calculate *category loadings* in cMCA. More precisely, category loadings, $\mathbf{L} \in \mathbb{R}^{K \times K'}$, under cMCA can be derived through:

$$\mathbf{L} = \mathbf{U} \text{diag}(\lambda)^{1/2} \tag{11}$$

which normalizes $\mathbf{U}$ with the corresponding *eigenvalues* (or *inertia*), $\lambda$. For negative eigenvalues, $\mathbf{L}$ is undefined. However, in practice, we only take a few top eigenvalues for analyses, and usually, we can expect that they are positive. Note that since Eq 11 derives only loadings for each category; thus, to know the influence of each variable (which contains a set of categories) on each contrastive PC, we need to obtain an aggregated measure of the loadings of each variable's categories. As an aggregated measure for each variable, we calculate a value range of

each variable's category loadings since the variable's influence on the separation of data points in their coordinates highly depends on the difference of the category loadings. For example, in a case where variable A has two categories with loadings of 0 and 2 and variable B has two categories with loadings of 1 and 2, while the total sum of loadings of B is larger than the one of A, A is more influential on making differences in data point coordinates. While we take a value range by default, one can use other statistical values based on their analysis focus (e.g., a variance of loadings when emphasizing more on outliers of the loadings is preferable).

## Application

Based on our substantive expertise, we analyze two national surveys—the CES 2020 and the ESS-UK 2018—to demonstrate the utility of cMCA while using the manual- or auto-selection of the hyperparameter $\alpha$.

As suggested by Le Roux and Rouanet [16], we recoded and restricted the number of categories/levels to deal with the potential issue caused by certain variables' active but *infrequent* categories, which could be overly influential in the determination of latent axes. Specifically, we first removed all missing values from each of the datasets; then, for variables that have more than five active levels, we pooled the adjacent two or three levels as one new level (For a more detailed recoding scheme, please refer to S6 Appendix). For instance, the left-right ideological scale in the ESS-UK 2018 is originally an eleven-point scale from zero through ten, we converted this variable as a new five-point scale: the original levels 0–1, 2–3, 4–6, 7–8, 9–10 are recoded as 1, 2, 3, 4, 5, respectively. Due to the focus of this paper being the usage of cMCA, we assume that the missing data is generated at random in our analyses; however, note that when the data is not missing at random, listwise deletion could cause analytical results to be inefficient and biased [40].

Given that the home countries of these surveys (i.e., the U.S. and the U.K.) have different political environments, these surveys provide a wide range of political situations for testing. Also, because the general political systems and cultures of each country are unique, we validate the consistency of the derived cMCA results with the observed, qualitative political realities in each country.

### Case one: CES 2020—The hidden conflicts regarding Trump and related issues

We firstly demonstrate the utility of cMCA by analyzing the 2020 Cooperative Election Study, conducted from September 29, 2020 to November 2, 2020 (the pre-election wave) and from November 8, 2021 to December 18, 2021 (the post-election wave). The principal investigators/ institutions, original data, and survey guide of CES 2020 can be retrieved from https:// dataverse.harvard.edu/dataset.xhtml?persistentId=doi%3A10.7910/DVN/E9N6PH. For this analysis, we focus on the pre-election wave and select 69 variables all related to national issues, including job approval, roll call votes, and salient policies and issues (e.g., abortion and gun control). For job approval questionnaires, those related to the president are only included in the analysis, given that other institutions are not solely run by a single nation-wise politician, and thus voters may lean more on local, rather than national, perspectives, to make evaluations. After deletion, the sample size reduces from 7,700 respondents to 5,616. In addition, to demonstrate the differences between analytical results of cMCA and ordinary scaling, we apply three different scaling methods to this data, each of which represents a different modeling strategy—MCA (nonparametric approach), blackbox scaling (parametric and frequentist approach), and OIRT (parametric and Bayesian approach)—and present the MCA result (Fig 1) in the main text and the rest in S1 Appendix (Figs 6 and 7) for references. Note that unlike

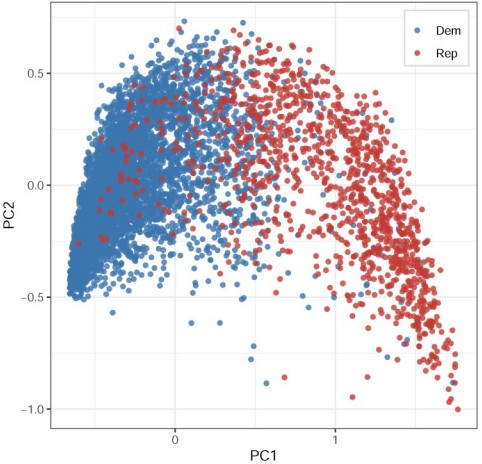

**Fig 1. MCA result of CES 2020.**

the blackbox function and MCA, OIRT does not estimate normal vectors of issues. Therefore, for the OIRT model results, we set the vector of `CC20.320a` as the *x*-axis and the vector of `ideo5` as the *y*-axis to approximate the latent pattern derived by the other two approaches (please refer to S6 Appendix for detailed coding schema).

Although the three ordinary scaling methods utilize different approaches for dimensional reduction and point estimation, their results show one similar pattern among American voters. Given that MCA does not derive "explained variance" for each variable but for each *category*, making a biplot-type figure may confuse readers, unlike the case for blackbox scaling. Instead, the category loadings plot and the category coordinates plot are provided in S2 Appendix. That is, the derived PC1 clearly splits American voters along with partisanship (`Dem`: Democrat or `Rep`: Republican), with some exceptions. For the sake of simplicity, we will mainly focus on the discussion of PC1 which explains the most of variation in the data in general. Nevertheless, auxiliary information related to PC2 is included for reference. Given that the positions are estimated through voters' self-reported values, these results demonstrate that the predispositions the U.S. voters heavily rely on are highly associated with their vote choices and party preferences [41, 42]. In general, although self-identified ideology is not the most prominent variable comprising PC1 (as seen in S2.1 in S2 Appendix), we find that PC1 is consistent with the liberal-conservative ideology as self-identified ideologies leaning toward liberal (responding 1 or 2 on the 5-point ideological scale), moderate (responding 3), and conservative (responding 4 or 5) are respectively observed on the left, center, and right sides of the figure. In other words, all ordinary scaling results demonstrate the existence of ideological polarization among the U.S. public, which is identified previously in the literature [43–46].

Furthermore, both the theoretical concept of ideology [47–49] and statistical interpretation of PCs [30, 50] are indeed defined as a (linear) combination of multiple issues/attitudes; thus, the scaling results of *variable vectors* help explore influential issues/variables and their magnitude of the influence on the composition of each PC. According to Table 1 in S2 Appendix, one can see that variables most contributed to PC1 are related to Trump's approval (i.e., `CC20.320a`, `CC20.350f`, and `CC20.350g`), border-related policies (`CC20.442c`), and nomination to the Supreme Court (`CC20.356`) during this time. As the validity check, in the category coordinates of each variable in Fig 8 in S2 Appendix, one can find that the categories

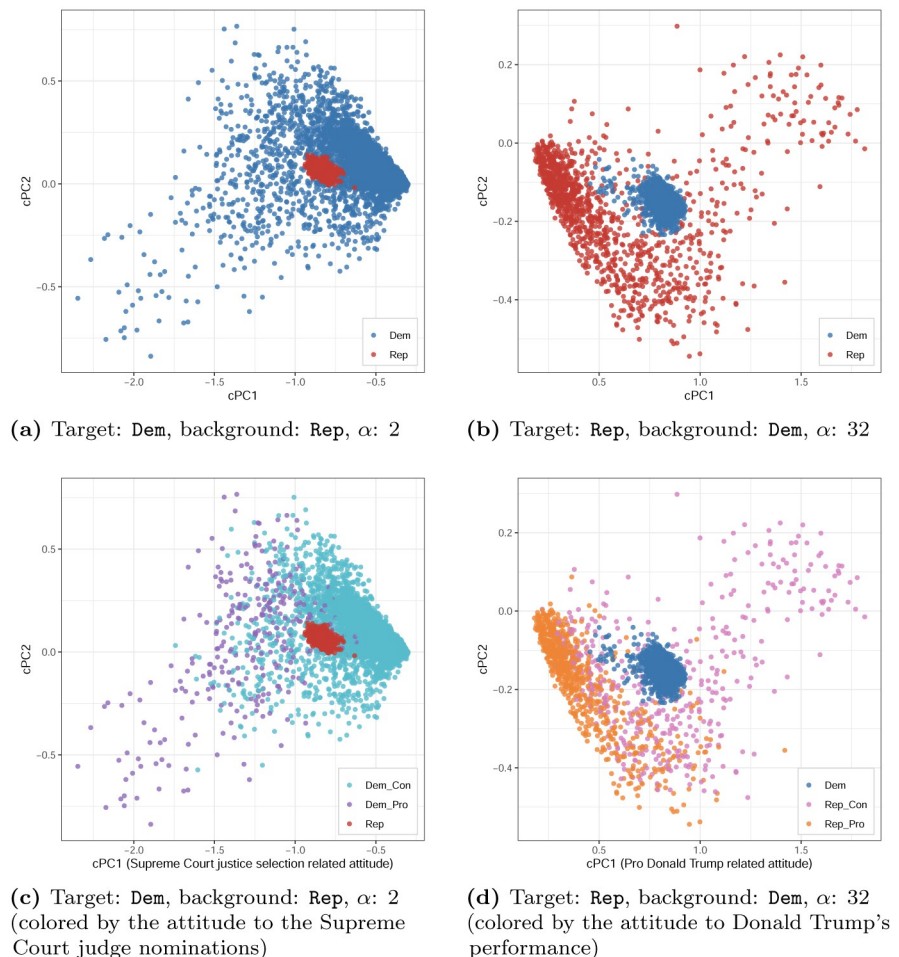

**(a)** Target: `Dem`, background: `Rep`, $\alpha$: 2

**(b)** Target: `Rep`, background: `Dem`, $\alpha$: 32

**(c)** Target: `Dem`, background: `Rep`, $\alpha$: 2 (colored by the attitude to the Supreme Court judge nominations)

**(d)** Target: `Rep`, background: `Dem`, $\alpha$: 32 (colored by the attitude to Donald Trump's performance)

**Fig 2. cMCA results of CES 2020.**

corresponding to liberal and conservative views on the aforementioned issues are also respectively placed on the left and right sides of the same space with the data-point coordinates.

To demonstrate how cMCA (with the manual-selection method) works differently from ordinary scaling, we generate cMCA results (i.e., data-point coordinates) shown in Fig 2a by assigning the Democrats to the target group and the Republicans to the background group. When $\alpha$ is *manually* set to 2, the Democrats (in blue color) can be divided by the Republicans (in red color) along the first contrastive PC (cPC1) and the second contrastive PC (cPC2) separately. This indicates that cMCA uncovers the directions along which the Democrats have much higher variations relative to the Republicans.

When using cMCA, we derive two types of auxiliary information—the *category coordinates* and *category loadings*—as heuristics to identify the most influential categories and variables. Please refer to S3 Appendix for this auxiliary information. As discussed earlier, a value range of each variable's category loadings guides us to select influential variables on the division or distance between respondents' positions along the latent direction. In addition, given that the category coordinates are in the same space as the data-point/respondent coordinates, a respondent who is placed near a certain category is likely to be associated with this category. Note that although one of the scaling method's main functions is to extract influential variables, this

does not necessarily mean that each derived PC simply corresponds to only a few influential variables; rather, each PC usually represents a linear combination of *all* variables [51]. For the sake of simplicity and clarity, in the following, we will only report and discuss the top few influential variables on cPC1 in the main text and provide a more detailed report in S5 Appendix.

Through the auxiliary information listed in S3.1 in S3 Appendix, we identify that the most influential variables/issues on the division of the Democrats on cPC1 are controversies over Trump's two most recent nominations to the Supreme Court. More precisely, the top-two issues (in order) that separate the Democrats along cPC1 are: Amy Coney Barrett's confirmation (`CC20.356`) and Brett Kavanaugh's confirmation (`CC20.350c`). Overall, the cMCA results indicate that the Democrats who hold liberal attitudes and disapprove of the two nominated Supreme Court judges are distributed to the right side of the space, whereas those who hold conservative views and approve of the two Supreme Court nominees are distributed on the left side of the space in Fig 2a.

We further color-encode the Democrats based on their approval of the nomination of the two new Supreme Court judges. Both variables are binary, where 1 refers to approval and 2 refers to disapproval. Please refer to S6 Appendix for more details. More precisely, the Democrats who disapprove of both judges are colored in `purple` (`Dem_Pro`) and the rest (i.e., those who approve of one or both judges) is colored in `teal` (`Dem_Con`) (see Fig 2c). By and large, Fig 2a and 2c conclude that compared with the other issues, these two issues together, the approval of Amy Coney Barrett and of Brett Kavanaugh, dominate the composition of cPC1 (i.e., the main variation), which divides the Democrats into two sub-groups.

On the other hand, cMCA also discovers a specific pattern within the Republicans when assigning Democrats to the background group. As presented in Fig 2b, when $\alpha$ is *manually* set to 32, we find that Republicans (`red`) are split by Democrats (`blue`) into two sub-groups along cPC1 and cPC2 separately. According to the auxiliary information in S3.1 in S3 Appendix, we identify that the top-two influential variables that compose cPC1 are all related to the approval of Trump's performance, i.e., the approval of Trump's job (`CC20.320a`) and whether to remove Trump due to abuse of power (`CC20.350f`). As seen in Fig 2d, cMCA uncovers a subgroup of Republicans who support and approve of Trump (`Rep_Pro` in `orange`), which differs from the one consisting of those who generally oppose him (`Rep_Con` in `pink`).

By and large, Fig 2 demonstrates that even when the PCs derived from the traditional methods are informative originally, cMCA can still identify influential variables and categories that create division within each of the predefined groups relative to the other. As demonstrated by the above results, although traditional scaling and cMCA may derive conceptually similar principal directions, the derived PCs may not necessarily be comprised of the same set of variables. Intuitively, the defined structure of the ideological scale within the general public (which includes both Republicans and Democrats) and solely within the Democrats or Republicans should be different. Indeed, as demonstrated, Trump's performance is the defining issue among the Republicans but only a partial issue for the general public. In other words, while there exists a general pattern across groups, cMCA provides an alternative way of exploring hidden patterns within each group that are different from the general pattern.

## Case two: ESS-UK 2018—The Brexit cleavage in the Labour and Conservative parties

The second survey we examine is the British module of the 2018 European Social Survey (ESS-UK 2018). We first manually select 23 variables that are generally related to political/

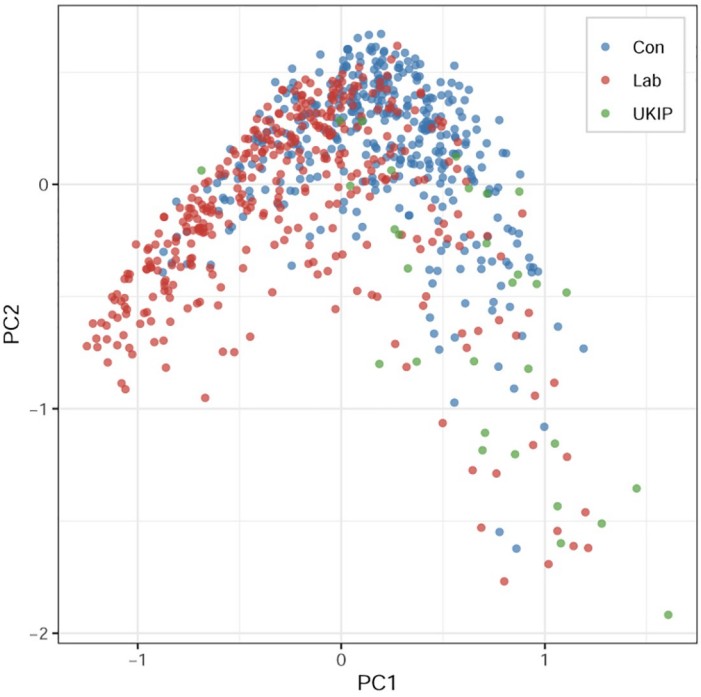

**Fig 3. MCA result of ESS-UK 2018.**

social issues and recode this data through the aforementioned procedure. For simplicity, we only include three partisans: the Conservative, Labour, and UK Independence Party (UKIP). After deletion, the sample size decreases from 946 respondents to 777. As in the case of CES 2020, we first apply all the aforementioned three ordinary scaling approaches to this survey. For the OIRT model of ESS, we set the vector of `lrscale` as the *x*-axis and the vector of `atcherp` as the *y*-axis. According to Fig 3 and S1.2 in S1 Appendix (for more detailed information, please refer to Table 2 and Fig 9 in S2 Appendix), one may find there is only a vague pattern that roughly divides British voters ideologically. These results demonstrate that different partisans highly overlap around the *area surrounding the origin*, with some respondents spread wider out. Roughly speaking, from right to left, the general positions of the different partisans in order along PC1 of the MCA result (Fig 3) are the UKIP, Conservative, and Labour supporters. However, there only exists a low level of political division among partisans in the U.K., unlike the case of American voters.

To contrast with standard scaling, we apply cMCA with the *automatic selection* of *α* to the same survey, and present the results in Figs 4 and 5. As shown in Fig 4, the first pair we examine includes the two major parties, the Conservative and Labour. According to Labour's positions in Fig 4a and the auxiliary information in S3.2 in S3 Appendix, when we specify the Labour as the target group, one can find that respondents' self-reported ideological position (`lrscale`) is the single most influential variable for both the first and second cPCs. Furthermore, based on cross-referencing the category and respondent coordinates, we see that Labour supporters could be further divided into several smaller groups along with their ideology from top-right (liberal leaning) toward bottom-left (conservative leaning) within the contrastive space (see Fig 4c). As shown in Fig 4e, we can further categorize these Labour supporters into three subgroups: those who are liberal (responses of 1, 2 to `lrscale`), moderate (a response of 3 to `lrscale`), and conservative (responses of 4 and 5 to `lrscale`).

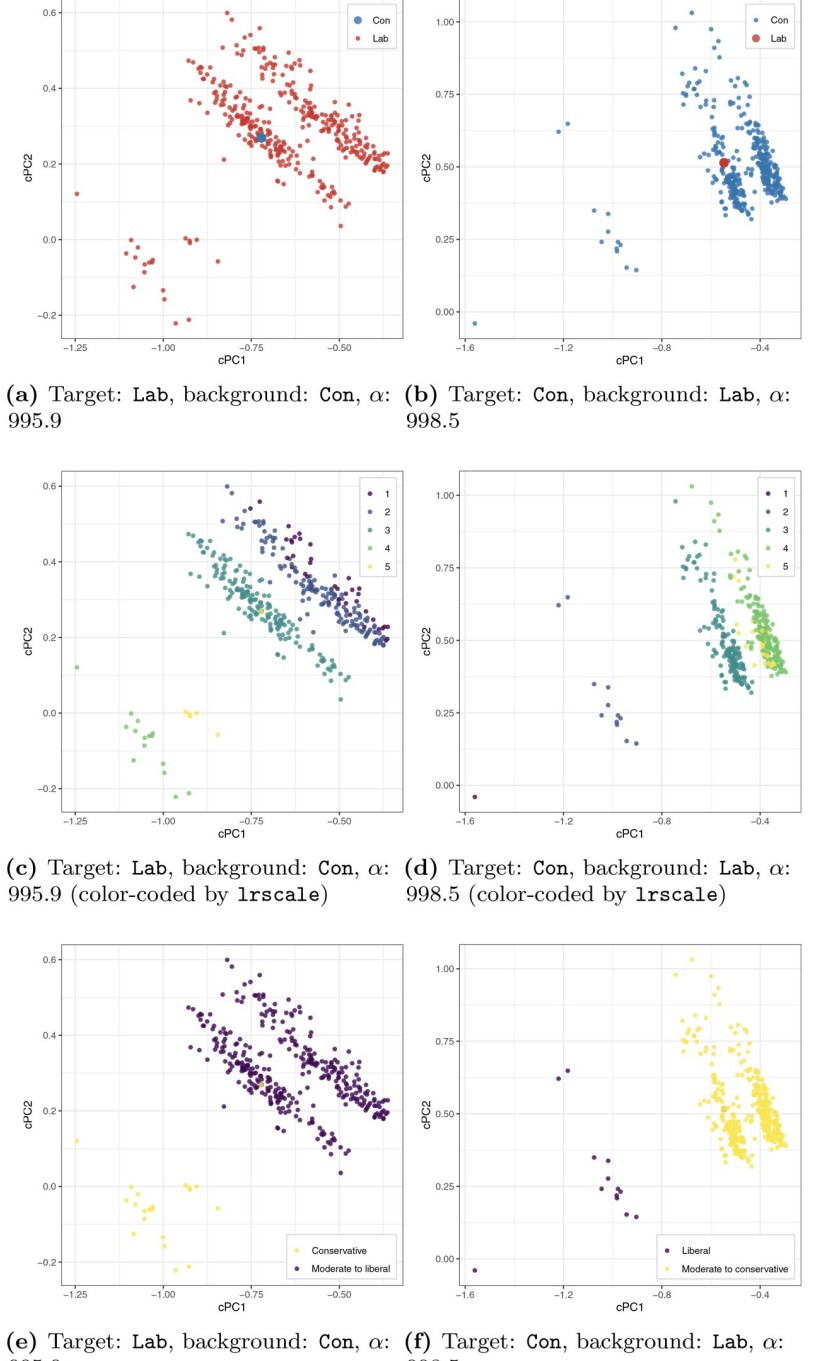

**(a)** Target: Lab, background: Con, $\alpha$: 995.9

**(b)** Target: Con, background: Lab, $\alpha$: 998.5

**(c)** Target: Lab, background: Con, $\alpha$: 995.9 (color-coded by `lrscale`)

**(d)** Target: Con, background: Lab, $\alpha$: 998.5 (color-coded by `lrscale`)

**(e)** Target: Lab, background: Con, $\alpha$: 995.9

**(f)** Target: Con, background: Lab, $\alpha$: 998.5

**Fig 4. cMCA results of ESS-UK 2018 (`Con` versus `Lab`).**

Similar to the above, when we take the Conservatives as the target group, we can see the respondents' self-reported ideological position is the single most influential variable again. The Conservatives also can be solely divided into several smaller groups along with their own ideology from the top-right (conservative leaning) towards the bottom-left (liberal leaning) within

the contrastive space (see Fig 4d). The groups can be further categorized into the Conservatives who are conservative, moderate, and liberal, as shown in Fig 4e.

By and large, by comparing Conservative and Labour supporters, the cMCA results show that ideology is the main differentiator between these two parties—the Labour (Conservative) supporters hold mostly moderate to liberal (conservative) opinions with some amount of contrarian outliers who hold views similar to their out-partisans. This result is greatly different than the results when applying traditional scaling to the entire dataset. On the one hand, the traditional scaling results such as Fig 3 and S1.2 in S1 Appendix demonstrate that British partisans are vaguely divided by ideology; on the other hand, cMCA explores the existing differences between two parties' ideological distributions, as depicted in Fig 4. Simply put, cMCA identifies ideology as the most effective variable to produce: 1) a high variance among Labour supporters relative to the Conservatives as liberal ideologies can be seen more among Labour supporters, and 2) a high variance among Conservatives relative to Labour as conservative ideologies can be seen more among Conservative supporters. Therefore, we conclude that cMCA finds that, in contrast to the American case, there exists a relatively low level of ideological polarization between Conservative and Labour supporters. This pattern of low-level polarization is obscured when using ordinary scaling because of the large degree of moderates found among Conservative and Labour supporters, about 43% and 45% separately. On the contrary, there are only about 37% and 23% of moderate supporters within the Democratic and Republican parties in the U.S. separately. Indeed, this recovered degree of low-level polarization is consistent with the current academic understanding that the U.K. has undergone a period of political depolarization since the second wave of ideological convergence between the elites of the Conservative and Labour parties [52–54].

In contrast, when cMCA contrasts each of the two main parties with UKIP, we can find different patterns, as shown in Fig 5. In Fig 5a where $\alpha$ is *automatically* set to around 1000, we observe that the Labour supporters (red) have much larger variance than the UKIP supporters (green). As shown in Eqs 5, 6 and 7, the automatic selection can reach a very large $\alpha$ value when there exist cPCs that produce an extremely large variance difference between the target and background groups. cMCA tends to be able to find such cPCs for the ESS-UK 2018 survey as the survey has a relatively large number of categories $K$ ($K = 200$) when compared to the number of rows/respondents of each group (Con: 383, Lab: 364, UKIP: 30). Consequently, the automatic selection sets $\alpha$ close to 1000, which is the upper bound of the search space using a default $\epsilon$ value ($\epsilon = 10^{-3}$). We set $\epsilon = 10^{-3}$ by default to follow the same upper bound of $\alpha$ suggested in the original cPCA [14]. However, when the number of rows/respondents is much larger than the number of categories, smaller $\epsilon$ values can be used (e.g., $\epsilon = 10^{-4}$) to extend the search space, and vice versa. According to S3.3 in S3 Appendix, the responses of 5 to imwbcnt (immigration makes the U.K. an *(extremely) better* place to live), the responses of 1 to imdfetn (allowing *many* immigrants of the different race as non-majority), and the responses of 5 to imueclt (the respondent's life is *extremely enriched* by immigrants) are the top-three influential categories, which "pull" some Labour supporters away from their co-partisans to the left side of cPC1 (all three variables are five-point scales—please refer to S6 Appendix for more details). Furthermore, from the category coordinates, we infer that the respondents who hold extreme attitudes are placed to the left side (e.g., 4 or 5 to imwbcnt). To illustrate the subgroups, as shown in Fig 5c, we color the Labour supporters who hold relatively EU-supportive opinions over all the three variables above with yellow (Lab_Pro) and the rest with pink (Lab_Oth). Based on these results, we find that Labour supporters who are on the left of cPC1 in Fig 5a are associated with relatively open views over these three immigration or Brexit-related variables.

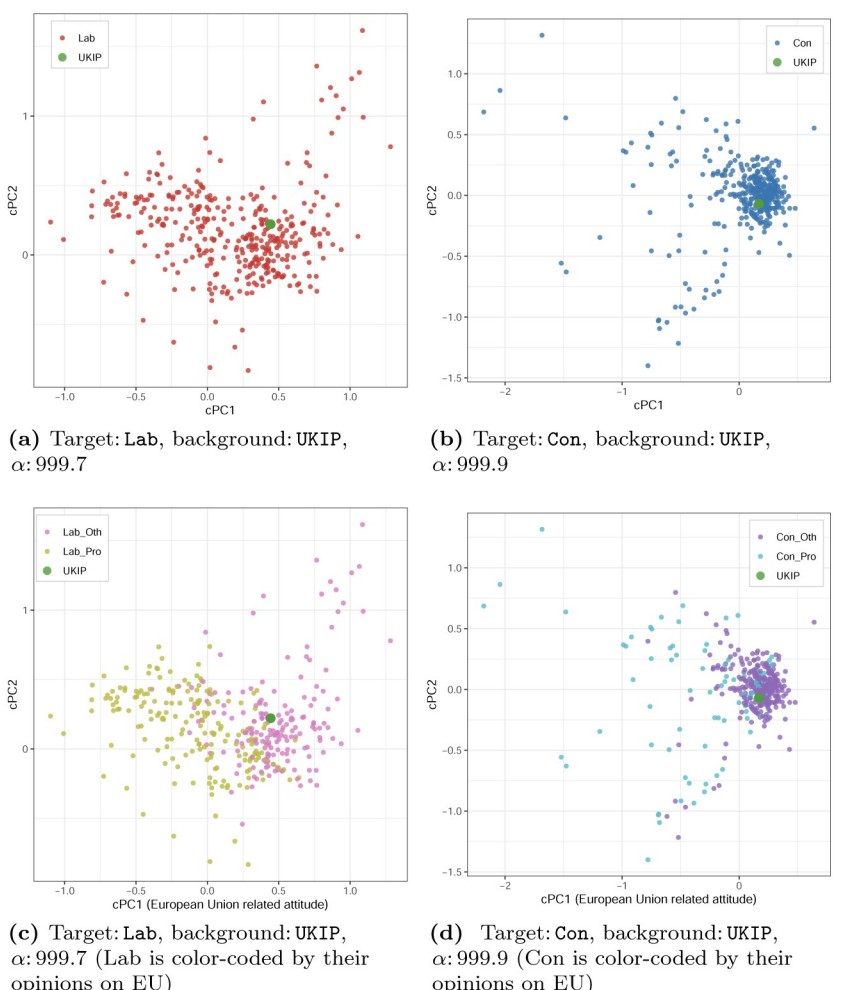

**(a)** Target: `Lab`, background: `UKIP`, $\alpha$: 999.7

**(b)** Target: `Con`, background: `UKIP`, $\alpha$: 999.9

**(c)** Target: `Lab`, background: `UKIP`, $\alpha$: 999.7 (Lab is color-coded by their opinions on EU)

**(d)** Target: `Con`, background: `UKIP`, $\alpha$: 999.9 (Con is color-coded by their opinions on EU)

**Fig 5. cMCA results of ESS-UK 2018 (`Lab` versus `UKIP`, `Con` versus `UKIP`).**

Analogously, we also find subgroups within the Conservatives by using the UKIP supporters as the background group. Similar to the previous results, as shown in Fig 5b, with an automatically selected $\alpha$ (around 1000), the Conservatives (`blue`) have much larger variance than the UKIP supporters (`green`). According to S3.3 in S3 Appendix, we see that the Conservative supporters located on the left side of cPC1 are also highly associated with the response of 5 to `imwbcnt`, `imueclt`, and `imbgeco` (the immigration makes the economy of the U.K. better). Similar to the dimension obtained for the Labour, we speculate that cPC1 represents a similar latent trait, which also divides the Conservative supporters into two subgroups—those who are relatively open on EU-related issues and the rest. In fact, as shown in Fig 5d, we can further divide the Conservative supporters into two sub-groups, where those who hold relatively open opinions over all three above variables are colored `teal` (Con_Pro) and the rest is colored `purple` (Con_Oth).

Overall, given that recent research has linked negative immigration attitudes to anti-EU sentiments [55], we conclude that both Conservative and Labour party supporters are internally divided by Brexit attitudes (with different magnitudes). As we may expect, although the two parties are divided by Brexit attitudes, the size of the pro-EU sub-group is significantly

larger among Labour supporters, as compared to Conservative ones: there are about 52% of the Labours who are in the subgroup of `Lab_Pro` and about 23% of the Conservatives who are in the subgroup of `Con_Pro`.

The results of Figs 4 and 5 demonstrate that although the derived variation from traditional scaling does not align with the boundaries between predefined groups (i.e., partisans), cMCA finds clear trends within those predefined groups. The results demonstrate the intrinsic differences in how traditional scaling and cMCA treat data, which lead to their divergent analytical outcomes. In addition, the ESS-UK 2018 analysis emphasizes an important component of cMCA—that is, the selection of the background group is highly influential in deriving results. Given that the basic concept of contrastive learning methods is to explore variation that is significant in the target group but insignificant in the background group, it is intuitive to see that the derived cPCs highly depend on one's selection. Consequently, one can imagine that if the selected target and background groups are highly similar, cMCA may not capture differences between two groups given the high level of similarity. Nevertheless, such "failure" can be also informative to know two groups' similarities. Thus, researchers can apply the contrastive learning method to any two groups to examine their level of similarity or dissimilarity.

## Discussion

Scaling has been widely used for both pattern recognition and latent-space derivation. Nevertheless, given that ordinary scaling methods only explore an overall latent pattern across groups, the derived results sometimes do not satisfy researchers whose interests instead focus on patterns *within* a group. In this article, we contribute to the scaling, contrastive learning, data mining, and data visualization literature by extending contrastive learning to MCA, enabling researchers to preserve analytical unbiasedness and efficiency as much as possible and to derive contrasted dimensions identifying subgroups hidden in data.

So far, while comparing latent patterns between multiple datasets, the main approach is to apply ordinary scaling first and compare the similarities or differences manually. However, this approach does not guarantee that the derived latent pattern is unique in the target group or the latent pattern considers the differences between the two groups. cMCA (or contrastive scaling in general) is designed to serve as a workhorse that simultaneously explores and compares multiple unique/different latent patterns. In short, this work demonstrates that cMCA, or contrastive methods in general, might provide novel insights overlooked by traditional methods when analyzing categorical survey data. To demonstrate this point, we apply MCA to only Democrats, Republicans, Labours, and Conservatives and provide category loadings of PC1 of each result in S4 Appendix as comparisons to the cMCA results in S3 Appendix. As one can see, given that the linear combination or the structure of PCs derived by different approaches are not the same (i.e., the sets of top-5 most influential variables derived by different approaches are not identical), this demonstrates that applying MCA to each of the two single groups separately does not generate the same results as applying cMCA to the two groups together. The structure of PCs generated by MCA is either totally different from those generated by cMCA (the Conservatives and Labours) or still similar to them but with more noise (the Democrats and Republicans).

Note that we are not implying that contrastive scaling is, in general, a replacement for previous methods. Our methodology falls into the category of other statistical and machine learning methods that were developed as new and unique approaches to tackle problems that a researcher may encounter. For example, local regression (e.g., LOESS or LOWESS) was developed specifically for a case where a researcher wants to reduce biases from parametric

assumptions. However, if a researcher is interested in having more generalizable results, ordinary least squares may be more useful. Similarly, we consider contrastive scaling as *not* a superior method for exploring multidimensional data, but rather an *alternative* that enables researchers to explore aspects of their data that are often missed by previous methods. Therefore, the question as to whether a researcher should adopt ordinary or contrastive scaling is one that solely depends on both the researchers' research question and own discretion—if researchers consider that there are no inter-group differences or are simply uninterested in exploring intra-group differences, they should utilize ordinary scaling; otherwise, contrastive scaling can be a powerful tool for exploring their data and answering their research questions.

It is also important to note that contrastive learning, including cMCA, is different from ordinary subgroup-analysis methods: Almost all standard methods for subgroup-analysis require researchers to have prior knowledge about how data should be "subgrouped," i.e., what variables/factors may cause differences between factions of an existing group. For example, as compared with two alternatives of MCA for subgroup analysis, class-specific MCA (CSA) and subgroup MCA (sMCA), cMCA allows researchers to *agnostically* explore all possible latent traits from different perspectives in the space. In other words, while conducting cMCA, without prior knowledge, researchers need only compare any two groups and apply different values of $\alpha$, either with manual- or auto-selection, to derive latent dimensions or traits which could subgroup data.

In contrast, both CSA and sMCA require researchers to *subjectively* subset/subgroup the original data along with certain traits first and then compare the derived patterns with the reference group, usually the original complete data. The ideas behind CSA and sMCA are similar: CSA is used to study whether a predefined subset of data points has a different latent pattern or not; sMCA is used to explore whether data points distribute differently after excluding certain subgroup(s) from the original data [16, 17]. Therefore, without any prior knowledge regarding how groups themselves could be divided, it is difficult for researchers to effectively apply CSA and sMCA to objectively explore subgroups. Especially, when data is *high-dimensional* (as in our two examples), this becomes more difficult because finding effective subgrouping criteria from many variables is not a trivial procedure. This comparison by no means indicates that cMCA is superior to CSA, sMCA, or other subgroup-analysis methods. Rather, it shows how cMCA is an additional tool for researchers to agnostically explore latent traits. In that vein, we believe that this new approach can assist to extract important features from high-dimensional data to define subgroups and complement existing subgroup-analysis methods.

There are several use cases for cMCA. First, cMCA can be used for analyzing covariate-balance between treatment and control groups in experiments. Indeed, cMCA can be utilized to explore the level of similarity of two sets of identical active variables—if cMCA finds a high variance only in one group, variables that contribute to the variance are likely to contain problems regarding the procedure of randomization. Second, cMCA can be applied to substantive topics. Recently, scholars of political polarization found that American voters' extremely ideological disagreement (i.e., ideological polarization), which usually aligns with partisan lines [56, 57], has formed solid in-group/out-group identities and further reinforced their emotional cleavage (i.e., affective polarization) [58–60]. However, for instance, given that the alignment between salient issue disagreement and partisan lines among British voters is not as clear as among American voters, a derived "issue cleavage" from cMCA, which could cause social distance among citizens and further create group/faction affiliation [61], can be a good source of studying affective polarization as well, in addition to party ID.

## Supporting information

**S1 Appendix. Results of ordinary scaling.** Blackbox scaling and ordinal item response theory model.
(PDF)

**S2 Appendix. Auxiliary information of MCA.** Category loadings and category coordinates.
(PDF)

**S3 Appendix. Auxiliary information of cMCA.** Category loadings and category coordinates.
(PDF)

**S4 Appendix. Applying MCA to single groups as comparisons.**
(PDF)

**S5 Appendix. Auxiliary information of MCA and cMCA in detail.**
(PDF)

**S6 Appendix. Variable coding scheme.**
(PDF)

## Author Contributions

**Writing – original draft:** Tzu-Ping Liu.

**Writing – review & editing:** Takanori Fujiwara.

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
