## [Decision Letter · Decision Letter 0]

24 Jan 2023

PONE-D-22-28924Contrastive Multiple Correspondence Analysis (cMCA): Using Contrastive Learning to Identify Latent Subgroups in Political PartiesPLOS ONE

Dear Dr. Liu,

Thank you for submitting your manuscript to PLOS ONE. After careful consideration, we feel that it has merit but does not fully meet PLOS ONE’s publication criteria as it currently stands. Therefore, we invite you to submit a revised version of the manuscript that addresses the points raised during the review process. I very much enjoyed reading your submission. I think all three readers are overall supportive but cite a variety of revisions that I think would make the article clearly for the reader. None, to my mind, necessitate major revisions but you are welcome to extend the due date should you need it. I think if you do the best to address all reviewer comments it will improve an already very good submission.

We look forward to receiving your revised manuscript.

Kind regards,

Lorien Shana Jasny

Academic Editor

PLOS ONE

Journal Requirements:

4. We note you have included a table to which you do not refer in the text of your manuscript. Please ensure that you refer to 

tables 4, 5, 6, 7 and 8in your text; if accepted, production will need this reference to link the reader to the Table

Reviewers' comments:

Reviewer's Responses to Questions

**Comments to the Author**

1. Is the manuscript technically sound, and do the data support the conclusions?

Reviewer #1: Yes

Reviewer #2: Partly

Reviewer #3: Yes

2. Has the statistical analysis been performed appropriately and rigorously? 

Reviewer #1: Yes

Reviewer #2: Yes

Reviewer #3: Yes

3. Have the authors made all data underlying the findings in their manuscript fully available?

Reviewer #1: Yes

Reviewer #2: Yes

Reviewer #3: Yes

4. Is the manuscript presented in an intelligible fashion and written in standard English?

Reviewer #1: Yes

Reviewer #2: Yes

Reviewer #3: Yes

5. Review Comments to the Author

Reviewer #1: This is an excellent and interesting paper that introduces an analysis technique for identifying subgroup latent patterns. I particularly value the clarity by which the methods and calculations are explained, which is comprehensive and easy to follow. The choice of context to apply the analysis also showcases its usefulness. I have the following comments and suggestions to help improve the paper before publication:

On page 8 you say the matrices must have the same dimensions, which I take to mean that the two groups for comparison must be the same size. Could you please a) clarify this and b) discuss whether the analysis can be performed on groups of different sizes. It seems that the requirement for symmetrical groups limits the application of this technique, or forces the researcher to make decisions about which observations to include in the groups in order to balance them. If this is the case, could you explain how to tackle asymmetrical groups should there be no way round it (i.e with some pre analysis).

Relatedly, your first step in preparing the data for application of the method involves removing missing cases. Could you please discuss what the consequences are (if any) for removing missingness.

A minor point but it is not immediately clear whether the CES and ESS are citizen surveys unless you are familiar with the data because in the earlier parts of the paper you talk about analysing Congress members. I recommend a sentence just clarifying this.

It would be useful to see in the main text a table or diagram of what is in the analysis that produces Figures 1 and 3, i.e. the substantive variables not just the variable codes as in the appendix. I would also recommend an extra sentence for each of the interpretations of these visualisations to make clear to all types of readers what is seen there, for instance "We can see this ideological polarisation because..."

If the main strength of this analysis is that it shows patterns/difference in variation, I wonder if you could more clearly describe or quantify this. For instance, 'Democrats are twice as similar to Republicans on the Supreme Court dimension as they are on Trump approval' or some kind of interpretation in the text like this.

Finally, another minor point but in the UK analysis (Figures 4 and 5) I would recommend making the background data point larger so it can clearly be seen. I am unsure why it is a large group in the US analysis but looks small in the UK -- perhaps this is due to N and this could be reported?

Reviewer #2: The authors propose a contrastive learning method developed expressly for categorical data, contrastive Multiple Correspondence Analysis (cMCA). cMCA is applied to two survey datasets, and its results are discussed.

Major Comments

• I’m a little puzzled by the limitations of traditional scaling methods outlined on pages 2 and 3. If researchers are seeking latent factors within sub-populations, like Democrats in the House of Representatives, instead of across a population, like the entirety of the House of Representatives, why not apply these scaling methods directly to the sub-population of interest? This should be addressed in the motivating example. Applying vanilla MCA to the subgroups in the analyses of Section 3, followed by a comparison of MCA’s latent factors to those produced by cMCA would also serve as an important demonstration of the proposed methods superiority over traditional methods.

• It’s stated that cMCA “objectively” and “agnostically” identifies subgroups in the last para- graph of page 4. Can you explain what is meant by these adverbs?

• Is it appropriate to claim that cMCA is objective given that analysts are encouraged to manually select the value of α that produces the most “meaningful” latent space? Wouldn’t this approach bring about spurious findings?

• Can you provide more information on the CES 2020 and ESS 2018 surveys for readers who might be unfamiliar with these data sources? In particular, how many observations and features are contained in each? How many respondents and questions were removed during preprocessing?

• I feel that the methodological contributions of this work are overstated. Just as MCA is PCA applied to data whose categorical variables are transformed into dummy variables, cMCA is cPCA applied to one-hot encoded data. cMCA is a less general form of cPCA that is limited to categorical data. I’d instead focus on the following: This work demonstrates that contrastive methods, like cPCA, might provide novel insights overlooked by traditional methods when analyzing categorical survey data.

Minor Comments

• Page 3, second paragraph: “Variation that is statistically small across groups may be signif- icantly large within groups, as seen in substantive applications in political science.” I don’t quite understand this sentence. What does it mean to for variation to be “statistically small”?

• Page 4, second paragraph: I believe that there’s a typo in “(due to variable-types cannot be analyzed)”.

• L is undefined when one of the K′ leading eigenvalues is negative. Perhaps this should be mentioned in the paragraph following Equation (11).

A pdf version of these comments are attached.

Reviewer #3: This is a well-written and highly innovative paper that combines two scaling methods and demonstrates their utility with two substantive examples. One of these scaling approaches (contrastive learning) is novel and stems largely from Abid et al.’s 2018 paper in Nature Communications. The other (Multiple Correspondence Analysis, or MCA) is a more established scaling technique, though it still underutilized and underappreciated in much of the social sciences, especially political science.

By combining the two, the paper provides an immensely valuable tool for social scientists looking to uncover heterogeneity in attitude structures. While much work has been done on developing methods for identifying heterogeneity in the context of treatment effects from experimental (and occasionally observational) data, very little effort has been put into extending these tools in the field of scaling methodology. Moreover, many of us in the scaling community have long felt that MCA should be more widely applied to political science problems, and I am happy to see this paper put it to good use here. Finally, I think the paper does a terrific job of walking the reader through these different components, explaining their mechanisms, and motivating their usage. The two substantive examples are relevant and effectively demonstrate the value of the contrastive MCA (cMCA) method in recovering richer sources of attitudinal heterogeneity among voters.

For all of these reasons, I am very excited to see this paper under review and hope to see it in print soon. My suggestions for revision are minor and stylistic:

1.) The first two paragraphs of the Introduction are a bit clunky. They divide scaling methods into two rough groups: “optimal scoring” (PCA, CA/MCA, and factor analysis) and “spatial voting” (NOMINATE, IRT, Optimal Classification, and others). This strikes me as an unconventional organization, especially because IRT models are a cumulative (vs. unfolding) scaling method that have been applied to measure and test spatial voting even though they are not a proximity model (as are unfolding methods like NOMINATE and OC). Admittedly many of these differences are technical/semantic. But I think that if the paper is going to use this framework, it should better explain the difference between the optimal scoring and spatial voting classes of methods. Jacoby’s (1991) Sage green book on Data Theory and Dimensional Analysis might provide a nice source.

2.) I think Poole’s “optimal classification” method should be capitalized.

3.) In the second paragraph of the Introduction, I’m not sure how accurate it is to list Coombs (1964) as the origin of the consistent finding of a left-right ideological organizing dimension. Certainly, Coombs’ work was hugely influential, but Coombs was a psychometrician and did very little work directly in political science. I think it would be more appropriate to cite the work that provided the foundation of what would become known as the “basic space theory of ideology” (Hinich and Munger 1994). The specific literature is varied, but some notable work includes Weisburg and Rusk (1970, “Dimensions of Candidate Evaluation”), Cahoon et al. (1978, “A Statistical Multidimensional Scaling Method Based on the Spatial Theory of Voting”), Rabinowitz (1973, “Spatial Models of Electoral Choice”), and of course Downs (1957).

4.) I think “U.S. Congresspersons” should be “U.S. Members of Congress (MCs)”. This is a more common term.

5.) I think the third paragraph of the Introduction could simply mention that traditional scaling methods only uncover certain kinds of intraparty divisions (e.g., the “Squad” or the Tea Party caucus) that are related to the general left-right dimension.

6.) On p. 4, I would change the first sentence in the final paragraph to “and the UK module of the 2018 European Social Survey (ESS-UK 2018)”.

7.) At some point (perhaps in Section 2.1 on p. 6), I think the paper should mention that MCA is a valuable tool for representing categorical variation in data but it has not been widely utilized in political science. However, there are exceptions such as Bonica (2014, “Mapping the Ideological Marketplace”); Gibson and Hare (2016, “Moral Epistemology and Ideological Conflict”); Blasius and Thiessen (2001, “Methodological Artifacts in Measures of Political Efficacy and Trust”), and perhaps others.

8.) Perhaps I’m being dense, but is there a parallel between the contrast parameter $\\alpha$ and the row/column scaling weight in biplots? If so, the authors might consider mentioning this parallel as a familiar point of comparison for the reader.

6. PLOS authors have the option to publish the peer review history of their article (what does this mean?). If published, this will include your full peer review and any attached files.

Reviewer #1: No

Reviewer #2: No

Reviewer #3: No

---

## [Author Response · Author response to Decision Letter 0]

11 Apr 2023

We made a response letter which includes all of the changes and responses for reviewers' conveniences. Please find the attached response letter.

---

## [Decision Letter · Decision Letter 1]

7 May 2023

PONE-D-22-28924R1Contrastive Multiple Correspondence Analysis (cMCA): Using Contrastive Learning to Identify Latent Subgroups in Political PartiesPLOS ONE

Dear Dr. Liu,

Thank you for submitting your manuscript to PLOS ONE. After careful consideration, we feel that it has merit but does not fully meet PLOS ONE’s publication criteria as it currently stands. Therefore, we invite you to submit a revised version of the manuscript that addresses the points raised during the review process.

I agree with both reviewers that the article has come a long way and will make a great contribution to the literature. However, I do agree with Reviewer 2 that more revisions are needed to make the piece as good as possible. I think you can attend to all of Reviewer 2's concerns within the scope of additional 'minor' revisions. Additionally, as they hint at but do not ask for directly, it would be useful to compare the cMCA results with MCA results if only Democrats were examined or only Republicans (to take your first example). This would show the value in using the comparative approach. This could either be in the main article or an appendix. I also request a bit more text in each appendix to describe a bit more what the reader is seeing.

We look forward to receiving your revised manuscript.

Kind regards,

Lorien Shana Jasny

Academic Editor

PLOS ONE

Journal Requirements:

Reviewers' comments:

Reviewer's Responses to Questions

**Comments to the Author**

1. If the authors have adequately addressed your comments raised in a previous round of review and you feel that this manuscript is now acceptable for publication, you may indicate that here to bypass the “Comments to the Author” section, enter your conflict of interest statement in the “Confidential to Editor” section, and submit your "Accept" recommendation.

Reviewer #1: All comments have been addressed

Reviewer #2: (No Response)

2. Is the manuscript technically sound, and do the data support the conclusions?

Reviewer #1: Yes

Reviewer #2: Partly

3. Has the statistical analysis been performed appropriately and rigorously? 

Reviewer #1: Yes

Reviewer #2: Yes

4. Have the authors made all data underlying the findings in their manuscript fully available?

Reviewer #1: Yes

Reviewer #2: Yes

5. Is the manuscript presented in an intelligible fashion and written in standard English?

Reviewer #1: Yes

Reviewer #2: Yes

6. Review Comments to the Author

Reviewer #1: (No Response)

Reviewer #2: Please see my attached comments. I can't upload them here because there is mathematical notation that must be formated in latex.

7. PLOS authors have the option to publish the peer review history of their article (what does this mean?). If published, this will include your full peer review and any attached files.

Reviewer #1: No

Reviewer #2: No

---

## [Author Response · Author response to Decision Letter 1]

23 May 2023

We have followed the editor's comment to add explanatory texts in each appendix. We have also followed Reviewer 2's comments to revise the manuscript and put all of our responses in the response letter.

---

## [Decision Letter · Decision Letter 2]

1 Jun 2023

Contrastive Multiple Correspondence Analysis (cMCA): Using Contrastive Learning to Identify Latent Subgroups in Political Parties

PONE-D-22-28924R2

Dear Dr. Liu,

We’re pleased to inform you that your manuscript has been judged scientifically suitable for publication and will be formally accepted for publication once it meets all outstanding technical requirements.

Kind regards,

Lorien Shana Jasny

Academic Editor

PLOS ONE

Additional Editor Comments (optional):

Reviewers' comments:

Reviewer's Responses to Questions

**Comments to the Author**

1. If the authors have adequately addressed your comments raised in a previous round of review and you feel that this manuscript is now acceptable for publication, you may indicate that here to bypass the “Comments to the Author” section, enter your conflict of interest statement in the “Confidential to Editor” section, and submit your "Accept" recommendation.

Reviewer #2: All comments have been addressed

2. Is the manuscript technically sound, and do the data support the conclusions?

Reviewer #2: Yes

3. Has the statistical analysis been performed appropriately and rigorously? 

Reviewer #2: Yes

4. Have the authors made all data underlying the findings in their manuscript fully available?

Reviewer #2: Yes

5. Is the manuscript presented in an intelligible fashion and written in standard English?

Reviewer #2: Yes

6. Review Comments to the Author

Reviewer #2: Thank you for addressing my comments. I believe the manuscript now provides much more support for your novel methodology. Note that a minor issue remains, but it should be very easy to address: The MCA results for Labours and Conservatives datasets are missing from appendix S4.

7. PLOS authors have the option to publish the peer review history of their article (what does this mean?). If published, this will include your full peer review and any attached files.

Reviewer #2: No

---

## [Editor Report · Acceptance letter]

26 Jun 2023

PONE-D-22-28924R2 

Contrastive multiple correspondence analysis (cMCA): using contrastive learning to identify latent subgroups in political parties 

Dear Dr. Liu:

I'm pleased to inform you that your manuscript has been deemed suitable for publication in PLOS ONE. Congratulations! Your manuscript is now with our production department. 

Kind regards, 

on behalf of

Dr. Lorien Shana Jasny 

Academic Editor

PLOS ONE